# Skip Connections Matter: On the Transferability of Adversarial Examples Generated with ResNets

**Dongxian Wu**[1,3]   **Yisen Wang**[2][†]   **Shu-Tao Xia**[1,3]   **James Bailey**[4]   **Xingjun Ma**[4]
[1]Tsinghua University
[2]Shanghai Jiao Tong University
[3]PCL Research Center of Networks and Communications, Peng Cheng Laboratory
[4]The University of Melbourne

## Abstract

Skip connections are an essential component of current state-of-the-art deep neural networks (DNNs) such as ResNet, WideResNet, DenseNet, and ResNeXt. Despite their huge success in building deeper and more powerful DNNs, we identify a surprising *security weakness* of skip connections in this paper. Use of skip connections *allows easier generation of highly transferable adversarial examples*. Specifically, in ResNet-like (with skip connections) neural networks, gradients can backpropagate through either skip connections or residual modules. We find that using more gradients from the skip connections rather than the residual modules according to a decay factor, allows one to craft adversarial examples with high transferability. Our method is termed *Skip Gradient Method* (SGM). We conduct comprehensive transfer attacks against state-of-the-art DNNs including ResNets, DenseNets, Inceptions, Inception-ResNet, Squeeze-and-Excitation Network (SENet) and robustly trained DNNs. We show that employing SGM on the gradient flow can greatly improve the transferability of crafted attacks in almost all cases. Furthermore, SGM can be easily combined with existing black-box attack techniques, and obtain high improvements over state-of-the-art transferability methods. Our findings not only motivate new research into the architectural vulnerability of DNNs, but also open up further challenges for the design of secure DNN architectures.

## 1 Introduction

In deep neural networks (DNNs), a skip connection builds a short-cut from a shallow layer to a deep layer by connecting the input of a convolutional block (also known as the residual module) directly to its output. While different layers of a neural network learn different "levels" of features, skip connections can help preserve low-level features and avoid performance degradation when adding more layers. This has been shown to be crucial for building very deep and powerful DNNs such as ResNet (He et al., 2016a;b), WideResNet (Zagoruyko & Komodakis, 2016), DenseNet (Huang et al., 2017) and ResNeXt (Xie et al., 2017). In the meantime, despite their superior performance, DNNs have been found extremely vulnerable to adversarial examples (or attacks), which are input examples slightly perturbed with an intention to fool the network to make a wrong prediction (Szegedy et al., 2013; Goodfellow et al., 2014; Ma et al., 2018; Bai et al., 2019; Wang et al., 2019; 2020). Adversarial examples often appear imperceptible to human observers, and are transferable across different models (Liu et al., 2017). This has raised security concerns on the deployment of DNNs in security critical scenarios, such as face recognition (Sharif et al., 2016), autonomous driving (Evtimov et al., 2018), video analysis (Jiang et al., 2019) and medical diagnosis (Ma et al., 2019).

Adversarial examples can be crafted following either a white-box setting (the adversary has full access to the target model) or a black-box setting (the adversary has no information of the target model). White-box methods such as Fast Gradient Sign Method (FGSM) (Goodfellow et al., 2014), Basic Iterative Method (BIM) (Kurakin et al., 2016), Projected Gradient Decent (PGD) (Madry et al., 2018) and Carlini and Wagner (CW) (Carlini & Wagner, 2017) often suffer from low transferability

---

[†]Correspondence to: Yisen Wang (eewangyisen@gmail.com)

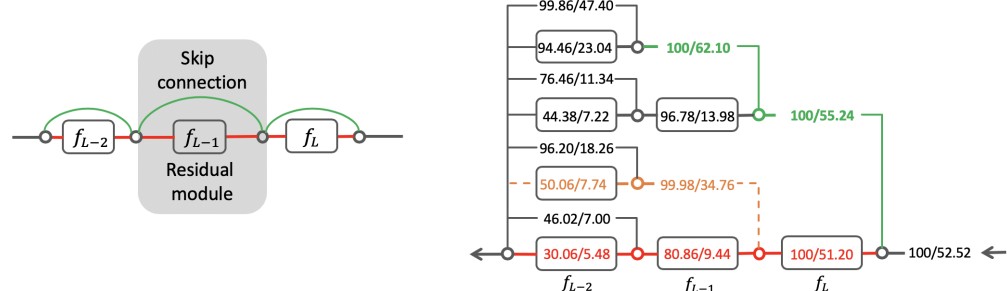

Figure 1: *Left*: Illustration of the last 3 skip connections (green lines) and residual modules (black boxes) of a ImageNet-trained ResNet-18. *Right*: The success rate (in the form of "white-box/black-box") of adversarial attacks crafted using gradients flowing through either a skip connection (going upwards) or a residual module (going leftwards) at each junction point (circle). Three example backpropagation paths are highlighted in different colors, with the green path skipping over the last two residual modules having the best attack success rate while the red path through all 3 residual modules having the worst attack success rate. The attacks are crafted by BIM on 5000 ImageNet validation images under maximum $L_\infty$ perturbation $\epsilon = 16$ (pixel values are in $[0, 255]$). The black-box success rate is tested against a VGG19 target model.

in a black-box setting, thus posing only limited threats to DNN models which are usually kept secret in practice (Dong et al., 2018; Xie et al., 2019). Several techniques have been proposed to improve the transferability of black-box attacks crafted on a surrogate model, such as momentum boosting (Dong et al., 2018), diverse input (Xie et al., 2019) and translation invariance (Dong et al., 2019). Although these techniques are effective, they (as well as white-box methods) all treat the entire network (either the target model or the surrogate model) as a single component while ignore its inner architectural characteristics. *The question of whether or not the DNN architecture itself can expose more transferability of adversarial attacks is an unexplored problem.*

In this paper, we identify one such weakness about the skip connections used by many state-of-the-art DNNs. We first conduct a toy experiment with the BIM attack and ResNet-18 on the ImageNet validation dataset (Deng et al., 2009) to investigate how skip connections affect the adversarial strength of attacks crafted on the network. At each of the last 3 skip connections and residual modules of ResNet-18, we illustrate the success rate of attacks crafted using gradients backpropagate through either the skip connection or the residual module in Figure 1. As can be observed, the success rate drops more drastically whenever using gradients from a residual module instead of the skip connection. This implies that gradients from the skip connections are more vulnerable (high success rate). In addition, we surprisingly find that skip connections expose more transferable information. For example, the black-box success rate was even improved from 52.52% to 62.10% when the attack skips the last two residual modules (following the path in green color).

Motivated by the above observations, in this paper, we propose the *Skip Gradient Method* (SGM) to generate adversarial examples using gradients more from the skip connections rather than the residual modules. In particular, SGM utilizes a decay factor to reduce gradients from the residual modules. We find that this simple adjustment on the gradient flow can generate highly transferable adversarial examples, and the more skip connections in a network, the more transferable are the crafted attacks. This is in sharp contrast to the design principles (*e.g.*, "going deeper" with skip connections) underpinning many modern DNNs. In particular, our main contributions are:

- We identify one surprising property of skip connections in ResNet-like neural networks, *i.e.*, they allow an easy generation of highly transferable adversarial examples.

- We propose the *Skip Gradient Method* (SGM) to craft adversarial examples using gradients more from the skip connections. Using a single decay factor on gradients, SGM is an appealingly simple and generic technique that can be used by any existing gradient-based attack methods.

- We provide comprehensive transfer attack experiments, from different source models against 10 state-of-the-art DNNs, showing that SGM can greatly improve the transferability of crafted adversarial examples. When combined with existing transfer techniques, SGM improves the state-of-the-art transferability benchmarks by a large margin.

## 2 RELATED WORK

Existing adversarial attacks can be categorized into two groups: 1) white-box attacks and 2) black-box attacks. In the white-box setting, the adversary has full access to the parameters of the target model, while in the black-box setting, the target model is kept secret from the adversary.

### 2.1 WHITE-BOX ATTACKS

Given a clean example $\boldsymbol{x}$ with class label $y$ and a target DNN model $f$, the goal of an adversary is to find an adversarial example $\boldsymbol{x}_{adv}$ that fools the network into making an incorrect prediction (eg. $f(\boldsymbol{x}_{adv}) \neq y$), while still remaining in the $\epsilon$-ball centered at $\boldsymbol{x}$ (eg. $\|\boldsymbol{x}_{adv} - \boldsymbol{x}\|_\infty \leq \epsilon$).

**Fast Gradient Sign Method (FGSM) (Goodfellow et al., 2014).** FGSM perturbs clean example $\boldsymbol{x}$ for one step by the amount of $\epsilon$ along the gradient direction:

$$\boldsymbol{x}_{adv} = \boldsymbol{x} + \epsilon \cdot \text{sign}(\nabla_{\boldsymbol{x}}\ell(f(\boldsymbol{x}), y)). \tag{1}$$

The Basic Iterative Method (BIM) (Kurakin et al., 2016) is an iterative version of FGSM that perturbs for $T$ steps with step size $\epsilon/T$.

**Projected Gradient Descent (PGD) (Madry et al., 2018).** PGD perturbs normal example $\boldsymbol{x}$ for $T$ steps with smaller step size. After each step of perturbation, PGD projects the adversarial example back onto the $\epsilon$-ball of $\boldsymbol{x}$, if it goes beyond the $\epsilon$-ball:

$$\boldsymbol{x}_{adv}^{t+1} = \Pi_\epsilon\big(\boldsymbol{x}_{adv}^t + \alpha \cdot \text{sign}(\nabla_{\boldsymbol{x}}\ell(f(\boldsymbol{x}_{adv}^t), y))\big), \tag{2}$$

where $\Pi_\epsilon(\cdot)$ is the projection operation. Different to BIM, PGD allows step size $\alpha > \epsilon/T$.

There are also other types of white-box attacks including sparsity-based methods such as Jacobian-based Saliency Map Attack (JSMA) (Papernot et al., 2016), sparse attack (Modas et al., 2019), one-pixel attack (Su et al., 2019), and optimization-based methods such as Carlini and Wagner (CW) (Carlini & Wagner, 2017) and elastic-net (EAD) (Chen et al., 2018).

### 2.2 BLACK-BOX ATTACKS

Black-box attacks can be generated by either attacking a surrogate model or using gradient estimation methods in combination with queries to the target model. Gradient estimation methods estimate the gradients of the target model using black-box optimization methods such as Finite Differences (FD) (Chen et al., 2017; Bhagoji et al., 2018) or Natural Evolution Strategies (NES) (Ilyas et al., 2018; Jiang et al., 2019). These methods all require a large number of queries to the target model, which not only reduces efficiency but also potentially exposes the attack. Alternatively, black-box adversarial examples can be crafted on a surrogate model then applied to attack the target model. Although the white-box methods can be directly applied on the surrogate model, they are far less effective in the black-box setting (Dong et al., 2018; Xie et al., 2019). Several transfer techniques have been proposed to improve the transferability of black-box attacks.

**Momentum Iterative boosting (MI) (Dong et al., 2018).** MI incorporates a momentum term into the gradient to boost the transferability:

$$\boldsymbol{x}_{adv}^{t+1} = \Pi_\epsilon\big(\boldsymbol{x}_{adv}^t + \alpha \cdot \text{sign}(\boldsymbol{g}^{t+1})\big), \ \boldsymbol{g}^{t+1} = \mu \cdot \boldsymbol{g}^t + \frac{\nabla_{\boldsymbol{x}}\ell(f(\boldsymbol{x}_{adv}^t), y)}{\|\nabla_{\boldsymbol{x}}\ell(f(\boldsymbol{x}_{adv}^t), y)\|_1}, \tag{3}$$

where $\boldsymbol{g}^t$ is the adversarial gradient at the $t$-th step, $\alpha = \epsilon/T$ is the step size for a total of $T$ steps, $\mu$ is a decay factor, and $\|\cdot\|_1$ is the $L_1$ norm.

**Diverse Input (DI) (Xie et al., 2019).** DI proposes to craft adversarial exampels using gradient with respect to the randomly-transformed input example:

$$\boldsymbol{x}_{adv}^{t+1} = \Pi_\epsilon\big(\boldsymbol{x}_{adv}^t + \alpha \cdot \text{sign}(\nabla_{\boldsymbol{x}}\ell(f(H(\boldsymbol{x}_{adv}^t; p)), y))\big), \tag{4}$$

where $H(\boldsymbol{x}_{adv}^t; p)$ is a stochastic transformation function on $\boldsymbol{x}_{adv}^t$ for a given probability $p$.

**Translation Invariant (TI) (Dong et al., 2019).** TI targets to evade robustly trained DNNs by generating adversarial examples that are less sensitive to the discriminative regions of the surrogate model. More specifically, TI computes the gradients with respect to a set of translated versions of the original input:

$$\boldsymbol{x}_{adv}^{t+1} = \Pi_\epsilon\big(\boldsymbol{x}_{adv}^t + \alpha \cdot \text{sign}(\boldsymbol{W} * \nabla_{\boldsymbol{x}}\ell(f(\boldsymbol{x}_{adv}^t), y))\big), \tag{5}$$

where $W$ is a predefined kernel (*e.g.*, uniform, linear, and Gaussian) matrix of size $(2k + 1)(2k + 1)$ ($k$ being the maximal number of pixels to shift). This kernel convolution is equivalent to the weighted sum of gradients over $(2k + 1)^2$ number of shifted input examples.

Furthermore, there are other studies focusing on intermediate feature representations. For example, Activation Attack (Inkawhich et al., 2019) drives the activation of a specified layer on a given image towards the layer of a target image, to yield a highly transferable targeted example. Intermediate Level Attack (Huang et al., 2019) attempts to fine-tune an existing adversarial example for greater black-box transferability by increasing its perturbation on a pre-specified layer of the source model.

Although the above transfer techniques are effective, they (including white-box attacks) either 1) treat the network (either the surrogate model or the target model) as a single component or 2) only use the intermediate layer output of the network. In other words, they do not directly consider the effects of different DNN architectural characteristics. Li et al. (2018) investigated the use of skip connections and dropout layers for sampling networks, which generates a huge set of ghost networks to perform an ensemble attack. Here, we focus on the architectural property of skip connections from the gradient view without modifying or generating any extra networks.

## 3 PROPOSED SKIP GRADIENT ATTACK

In this section, we first introduce the gradient decomposition of skip connection and residual module. Following that, we propose our Skip Gradient Method (SGM), then demonstrate the adversarial transferability property of skip connection via a case study.

### 3.1 GRADIENT DECOMPOSITION WITH SKIP CONNECTIONS

In ResNet-like neural networks, a skip connection uses identity mapping to bypass residual layers, allowing data flow from a shallow layer directly to subsequent deep layers. Thus, we can decompose the network into a collection of paths of different lengths (Veit et al., 2016). We denote a skip connection together with its associated residual module as a building block (residual block) of a network. Considering three successive building blocks (eg. $z_{i+1} = z_i + f_{i+1}(z_i)$) in a residual network from input $z_0$ to output $z_3$, the output $z_3$ can be expanded as:

$$z_3 = z_2 + f_3(z_2) = [z_1 + f_2(z_1)] + f_3(z_1 + f_2(z_1))$$
$$= [z_0 + f_1(z_0) + f_2(z_0 + f_1(z_0))] + f_3\big((z_0 + f_1(z_0)) + f_2(z_0 + f_1(z_0))\big). \tag{6}$$

According to the chain rule in calculus, the gradient of a loss function $\ell$ with respect to input $z_0$ can then be decomposed as,

$$\frac{\partial \ell}{\partial z_0} = \frac{\partial \ell}{\partial z_3}\frac{\partial z_3}{\partial z_2}\frac{\partial z_2}{\partial z_1}\frac{\partial z_1}{\partial z_0} = \frac{\partial \ell}{\partial z_3}(1 + \frac{\partial f_3}{\partial z_2})(1 + \frac{\partial f_2}{\partial z_1})(1 + \frac{\partial f_1}{\partial z_0}). \tag{7}$$

Extending this toy example to a network with $L$ residual blocks, the gradient can be decomposed from $L$-th to the $(l + 1)$-th ($0 \leq l < L$) residual block as,

$$\frac{\partial \ell}{\partial x} = \frac{\partial \ell}{\partial z_L} \prod_{i=l}^{L-1} \big(\frac{\partial f_{i+1}}{\partial z_i} + 1\big)\frac{\partial z_l}{\partial x}. \tag{8}$$

The example illustrated in Figure 1 is a the above decomposition of a ResNet-18 at the last 3 building blocks ($l = L - 3$).

### 3.2 SKIP GRADIENT METHOD (SGM)

In order to use more gradient from the skip connections, here, we introduce a decay parameter into the decomposed gradient to reduce the gradient from the residual modules. Following the decomposition in Equation (8), the "skipped" gradient is,

$$\nabla_x \ell = \frac{\partial \ell}{\partial z_L} \prod_{i=0}^{L-1} \big(\gamma \frac{\partial f_{i+1}}{\partial z_i} + 1\big)\frac{\partial z_0}{\partial x}, \tag{9}$$

where $z_0 = x$ is the input of the network, and $\gamma \in (0, 1]$ is the decay parameter. Accordingly, given a clean example $x$ and a DNN model $f$, an adversarial example can be crafted iteratively by,

$$x_{adv}^{t+1} = \Pi_\epsilon\Big(x_{adv}^t + \alpha \cdot \text{sign}\big(\frac{\partial \ell}{\partial z_L} \prod_{i=0}^{L-1} (\gamma \frac{\partial f_{i+1}}{\partial z_i} + 1)\frac{\partial z_0}{\partial x}\big)\Big). \tag{10}$$

Table 1: The success rates (%±std over 5 random runs) of black-box attacks (untargeted) crafted by PGD and its "skip gradient" (SGM) version, on different source models against a Inception V3 target model. The best results are in **bold**.

| | RN18 | RN34 | RN50 | RN101 | RN152 | DN121 | DN169 | DN201 |
|---|---|---|---|---|---|---|---|---|
| PGD | 23.23±0.69 | 24.38±0.41 | 22.80±0.55 | 22.98±0.83 | 26.56±0.75 | 30.71±0.60 | 30.90±0.31 | 36.01±0.59 |
| SGM | **28.92±0.45** | **43.43±0.32** | **36.71±0.55** | **38.38±0.53** | **44.84±0.14** | **57.38±0.14** | **60.45±0.42** | **65.48±0.23** |

SGM is a generic technique that can be easily implemented on any neural network that has skip connections. During the backpropagation process, SGM simply multiplies the decay parameter to the gradient whenever it passes a residual module. Therefore, SGM does not require any computation overhead, and works efficiently even on densely connected networks such as DenseNets. The reduction of residual gradients is accumulated along the backpropagation path, that is, the residual gradients at lower layers will be reduced more times than those at higher layers. This is because, compared to high-level features, low-level features have already been well preserved by skip connections (see feature decompositions in Equation (6)).

### 3.3 ADVERSARIAL TRANSFERABILITY WITH SKIP CONNECTIONS: A CASE STUDY

To demonstrate the adversarial transferability of skip connections, we conduct a case study on 10-step PGD, and their corresponding SGM versions, to investigate the success rates of black-box attacks crafted with or without manipulating the skip connections. The black-box attacks are generated on 8 different source (surrogate) models ResNet(RN)-18/34/50/101/152 and DenseNet(DN)-121/169/201, then applied to attack a Inception V3 target model. All models were trained on ImageNet training set. We randomly select 5000 ImageNet validation images that are correctly classified by all source models, and craft untargeted attacks under maximum $L_\infty$ perturbation $\epsilon = 16$, which is a typical black-box setting (Dong et al., 2018; Xie et al., 2019; Dong et al., 2019). The step size of PGD was set to $\alpha = 2$, and the decay parameter of SGM was set to $\gamma = 0.5$.

We run the attack for 5 times with different random seeds, and report the success rates (transferability) of different methods in Table 1. As can be seen, when the skip connections are manipulated with our SGM, the transferability of PGD is greatly improved across all source models. On all source models except RN18, the improvements are more than 13%. Without SGM, the best transferability against the Inception-V3 target model is 35.48% which is achieved by PGD on DN201, however, this is improved further by our proposed SGM to 65.38% ( > 29% gain). This not only highlights the surprising property of skip connections in terms of the generation of highly transferable attacks, but also indicates the significance of this property, as such a huge boost in transferability only takes a single decay factor.

The 8 source models can be interpreted as from 3 ResNet families: 1) RN18/34 are ResNets with normal residual blocks, 2) RN50/101/152 are ResNets with "bottleneck" residual blocks, and 3) DN121/169/201 are densely connected ResNets. Another important observation is that when there are more skip connections in a network within the same ResNet family (*e.g.*, RN34 > RN18, RN152 > RN101 > RN50, and DN201 > DN169 > DN121), or from ResNets to DenseNets (*e.g.* DN121/169/201 > RN18/34 and DN121/169/201 > RN50/101/152), the crafted adversarial examples become more transferable, especially when the skip connections are manipulated by our SGM. This raises questions about the design principle behind many state-of-the-art DNNs: "going deeper" with techniques like skip connection and $1 \times 1$ convolution.

## 4 COMPARISON TO EXISTING TRANSFER ATTACKS

In this section, we compare the transferability of adversarial examples crafted by our proposed SGM and existing methods on ImageNet against both unsecured and secured target models.

**Baselines.** We compare SGM with FGSM, PGD, and 3 state-of-the-art transfer attacks: (1) Momentum Iterative (MI) (Dong et al., 2018), (2) Diverse Input (DI) (Xie et al., 2019), and (3) Transition Invariant (TI) (Dong et al., 2019). Note that the TI attack was originally proposed to attack secured models, although here we include TI to attack both unsecured models and secured models. For TI and our SGM, we test both the one-step and the iterative version, however, the other methods DI and MI only have an iterative version. The iteration step is set to 10 and 20 for unsecured and secured target models respectively. For all iterative methods PGD, TI and our SGM, the step size is set to $\alpha = 2$. For our proposed SGM, the decay parameter is set to $\gamma = 0.2$ (0.5) and $\gamma = 0.5$ (0.7)

on ResNet and DenseNet source models in PGD (FGSM) respectively. For simplicity, we utilize SGM to indicate FGSM+SGM in one-step attacks, and PGD+SGM in multi-step attacks. Other parameters of existing methods are configured as in their original papers.

**Threat Model.** We adopt a black-box threat model in which adversarial examples are generated by attacking a source model and then applied to attack the target model. The target model is of a different architecture (indicated by the model name) to the source model, expect when the source and target models are of the same architecture, where we directly use the source model as the target model (equivalent to a white-box setting). The attacks are crafted on 5000 randomly selected ImageNet validation images that are classified correctly by all source models, and are repeated for 5 times with different random seeds. For all attack methods, we follow the standard setting (Dong et al., 2018; Xie et al., 2019) to craft untargeted attacks under maximum $L_\infty$ perturbation $\epsilon = 16$ with respect to pixel values in $[0, 255]$.

**Target Models.** We consider two types of target models: 1) unsecured models that are trained on ImageNet training set using traditional training; and 2) secured models trained using adversarial training. For unsecured target model, we choose 7 state-of-the-art DNNs: VGG19 (with batch normalization) (Simonyan & Zisserman, 2015), ResNet-152 (RN152) (He et al., 2016a), DenseNet-201 (DN152), 154 layer Squeeze-and-Excitation network (SE154) (Hu et al., 2018), Inception V3 (IncV3) (Szegedy et al., 2016), Inception V4 (IncV3) (Szegedy et al., 2017) and Inception-ResNet V2 (IncResV2) (Szegedy et al., 2017). For secured target models, we consider 3 robustly trained DNNs using ensemble adversarial training (Tramèr et al., 2018): $IncV3_{ens3}$ (ensemble of 3 IncV3 networks), $IncV3_{ens4}$ (ensemble of 4 IncV3 networks) and $IncResV2_{ens3}$ (ensemble of 3 IncResV2 networks).

**Source Models.** We choose 8 different source models from the ResNet family: ResNet(RN)-18/34/50/101/152 and DenseNet(DN)-121/169/201. Whenever the input size of the source model does not match the target model, we resize the crafted adversarial images to the input size of the target model. For VGG19, ResNet and DenseNet models, images are cropped and resized to $224 \times 224$, while for Inception/Inception-ResNet models, images are cropped and resized to $299 \times 299$.

### 4.1 TRANSFERABILITY AGAINST UNSECURED MODELS

We first investigate the transferability of all attack methods against 7 unsecured models, which is to find the best method that can generate the most transferable attacks on *one* source model against *all* target models.

**One-step Transferability.** The one-step transferability is measured by the success rate of one-step attacks, as reported in Table 2. Here, we only show the results on two source models: 1) RN152 which is the best ResNet source model with the highest success rate on average against all target models, and 2) DN201 which is the best DenseNet source model. Also note that, when the source and target models are the same, the result represents the white-box success rate. Overall, adversarial examples crafted on DN201 have significantly better transferability than those crafted on RN152, especially for our SGM method. This is because there are $\sim 30\times$ more skip connections that can be manipulated by our SGM in DN201 compared to RN152. In comparison to to both FGSM and TI, transferability is improved considerably by SGM in almost all test scenarios, except when transferring from RN152 to VGG19/IncV3/IncV4 where SGM is outperformed by TI. This implies that, when transfereing across different architectures (eg. ResNet $\rightarrow$ VGG/Inception), translation adaptation may help increase the transferability of one-step perturbations. However, this advantage of TI disappears when there are more skip connections, as is the case for the DN201 source model.

Table 2: One-step transferability: the success rates (%±std over 5 random runs) of black-box attacks crafted by different methods on 2 source models against 7 *unsecured* target models. The best results are in **bold**.

| Source | Attack | VGG19 | RN152 | DN201 | SE154 | IncV3 | IncV4 | IncResV2 |
|---|---|---|---|---|---|---|---|---|
| RN152 | FGSM | 41.96±0.52 | 71.53±0.34 | 37.49±0.10 | 30.00±0.56 | 25.66±0.07 | 21.55±0.16 | 19.90±0.49 |
| | TI | **49.61±0.11** | 49.33±0.35 | 36.87±0.42 | 29.95±0.32 | **33.59±0.73** | **29.05±0.34** | 20.62±0.09 |
| | **SGM** | 47.54±0.14 | **76.90±0.60** | **43.73±0.21** | **31.16±0.45** | 29.41±0.24 | 25.11±0.20 | **22.63±0.15** |
| DN201 | FGSM | 49.87±0.17 | 38.89±0.29 | 81.51±0.33 | 34.94±0.53 | 31.21±0.47 | 27.08±0.23 | 23.87±0.45 |
| | TI | 54.37±0.58 | 33.49±0.18 | 57.71±0.05 | 34.46±0.47 | 34.45±0.25 | 30.17±0.23 | 20.36±0.33 |
| | **SGM** | **56.97±0.25** | **47.54±0.14** | **87.73±0.76** | **42.31±0.67** | **37.91±0.56** | **32.83±0.38** | **29.64±0.25** |

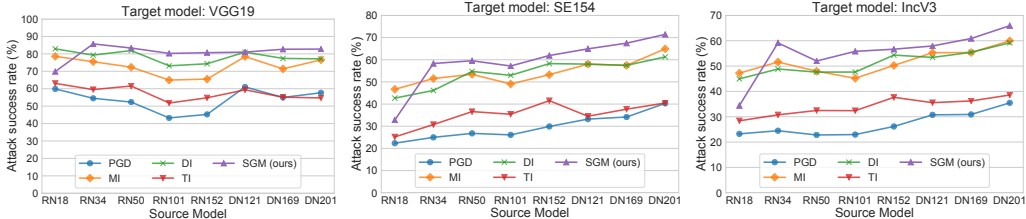

Figure 2: The attack success rates of black-box attacks crafted by different attack methods on 8 source models against 3 *unsecured* target models: VGG19 (left), SE154 (middle) and IncV3 (right).

Table 3: Multi-step transferability: the success rates (%±std over 5 random runs) of black-box attacks crafted by different methods on 2 source models against 7 *unsecured* target models. The best results are in **bold**.

| Source | Attack | VGG19 | RN152 | DN201 | SE154 | IncV3 | IncV4 | IncRes |
|--------|--------|-------|-------|-------|-------|-------|-------|--------|
| RN152 | PGD | 45.03±0.21 | **99.91±0.04** | 51.49±0.51 | 29.35±0.49 | 26.56±0.75 | 21.03±0.19 | 19.10±0.37 |
| | TI | 54.59±0.63 | 99.73±0.11 | 63.77±0.93 | 41.89±0.58 | 37.97±0.39 | 36.25±1.13 | 28.90±0.52 |
| | MI | 65.42±0.60 | 99.77±0.02 | 75.79±0.69 | 53.07±0.44 | 50.22±0.07 | 43.32±0.27 | 41.71±0.36 |
| | DI | 74.01±0.48 | 99.90±0.02 | 77.81±0.80 | 57.49±1.22 | 53.95±0.68 | 47.16±0.52 | 43.47±0.30 |
| | **SGM** | **79.90±0.69** | 99.87±0.03 | **81.56±0.35** | **61.83±0.17** | **57.22±0.51** | **48.57±0.09** | **45.44±0.36** |
| DN201 | PGD | 57.61±0.82 | 59.84±0.76 | **99.89±0.02** | 39.78±0.69 | 36.01±0.59 | 31.76±0.27 | 25.92±0.13 |
| | TI | 54.90±0.75 | 50.63±1.02 | 99.64±0.07 | 40.40±0.31 | 39.13±0.52 | 37.03±1.00 | 28.83±0.49 |
| | MI | 75.09±0.80 | 76.39±0.61 | 99.84±0.05 | 64.38±0.69 | 59.62±0.36 | 54.85±0.56 | 50.05±0.25 |
| | DI | 78.11±0.56 | 78.18±0.91 | 99.81±0.05 | 61.75±0.88 | 60.04±0.81 | 56.15±0.36 | 49.00±0.83 |
| | **SGM** | **82.66±0.29** | **86.65±0.50** | 99.67±0.08 | **72.03±0.53** | **65.48±0.23** | **58.77±0.78** | **54.97±0.25** |

**Multi-step Transferability.** First we provide a detailed study about the transferability of all attack methods from the 8 source models to the 3 representative unsecured target models. We then compare different attack methods on two best source models against all unsecured target models: the best ResNet source model and the best DenseNet source model. The multi-step (*e.g.*, 10 step) transferability from all source models to three representative target models (VGG19, SE154 and IncV3) is illustrated in Figure 2. In all transfer scenarios, our proposed SGM outperforms existing methods consistently on almost all source models except RN18. Adversarial attacks crafted by SGM become more transferable when there are more skip connections in the source model (*e.g.*, from RN18 to DN201). An interesting observation is that, when the target model is shallow such as VGG19 (left figure in Figure 2), shallow source models transfer better, however, when the target model is deep such as SE154 and IncV3 (middle and right figures in Figure 2), deeper source models tend to have better transferability. We suspect this is due to the architectural similarities shared by the target and source models. Note that against the VGG19 target model, the success rate of baseline methods all drop significantly when the ResNet source models become more complex (from RN18 to RN152). The small variations at RN50 and DN121 source models may be caused by the architectural difference between RN18/34 which consist of normal residual blocks, RN50/101/152 which consist of "bottleneck" residual blocks and DN121/169/201 which has dense skip connections.

Results for the best source models RN152 and DN201 against unsecured target models are reported in Table 3. The proposed SGM attack outperforms existing methods by a large margin consistently against different target models. Particularly, for transfer DN201 → SE154 (a recent state-of-the-art DNN with only 2.251% top-5 error on ImageNet), SGM achieves a success rate of 72.03%, which is > 7% and > 10% higher than MI and DI respectively.

**Combining with Existing Methods.** We further demonstrate that the adversarial transferability of skip connections can be exploited in combination with existing techniques. The experiments are conducted on DN201 (the best source model in the above multi-step experiments), and TI attack is excluded as it was originally proposed against secured models and demonstrates limited improvement over PGD against unsecured models. The results are reported in Table 4*. The transferability of MI and DI is improved remarkably of 11.98% ∼ 21.98% by SGM. When combined with both MI

---

*For simplicity, we omit the standard deviations here as they are very small, which hardly effect the results.

Table 4: Combined with existing methods: the success rates (%) of attacks crafted on source model DN201 against 7 *unsecured* target models. The best results are in **bold** and **+** indicates improvement.

| Attack \Target | VGG19 | RN152 | DN201 | SE154 | IncV3 | IncV4 | IncRes |
|---|---|---|---|---|---|---|---|
| MI | 75.09 | 76.39 | **99.84** | 64.38 | 59.62 | 54.85 | 50.05 |
| MI+SGM | +12.01 | +13.24 | 99.52 | +17.16 | +21.88 | +15.57 | +18.35 |
| DI | 78.11 | 78.18 | 99.81 | 61.75 | 60.04 | 56.15 | 49.00 |
| DI+SGM | +12.28 | +13.76 | 99.52 | +20.92 | +17.66 | +15.78 | +20.20 |
| MI+DI | 87.16 | 87.28 | 99.76 | 79.80 | 76.68 | 75.20 | 71.05 |
| **MI+DI+SGM** | **93.00** | **93.92** | 99.42 | **89.86** | **85.72** | **81.23** | **80.50** |

Table 5: Transferability against secured models: the success rates (%±std over 5 random runs) of multi-step attacks crafted on RN152 and DN201 source models against 3 secured models. The best results are in **bold**.

| Source | Attack | IncV3$_{ens3}$ | IncV$_{ens4}$ | IncRes$_{ens3}$ |
|---|---|---|---|---|
| RN152 | PGD | 12.47±1.27 | 10.72±1.37 | 6.97±0.71 |
| | TI | **45.36±0.97** | **45.81±0.93** | **38.19±0.81** |
| | MI | 24.20±1.15 | 22.04±0.98 | 16.10±0.56 |
| | DI | 28.48±1.21 | 24.19±1.22 | 17.31±0.77 |
| | SGM | 31.57±0.55 | 27.77±0.47 | 20.02±0.66 |
| | **TI+SGM** | **52.62±0.40** | **52.80±0.79** | **43.96±0.62** |
| DN201 | PGD | 18.16±0.56 | 15.30±0.62 | 10.40±0.49 |
| | TI | **42.76±0.91** | **42.01±0.79** | **34.28±0.88** |
| | MI | 31.79±0.83 | 28.21±0.15 | 20.60±0.38 |
| | DI | 34.84±1.35 | 29.23±0.83 | 21.64±0.80 |
| | SGM | 41.45±0.30 | 37.85±0.22 | 29.41±0.02 |
| | **TI+SGM** | **46.11±1.23** | **47.38±0.89** | **39.32±0.80** |

and DI, SGM improves the state-of-the-art (MI+DI) transferability by a huge margin consistently against all target models. In particular, SGM pushes the new state-of-the-art to at least 80.52% which previously was only 71%. This illustrates that skip connections can be easily manipulated to craft highly transferable attacks against many state-of-the-art DNN models.

## 4.2 TRANSFERABILITY AGAINST ROBUSTLY TRAINED MODELS

The success rates of our SGM and other baseline methods against the 3 secured target models are reported in Table 5. Overall, with translation adaptation specifically designed for evading adversarially trained models, TI achieves the best standalone transferability, while SGM is the second best with higher success rates than either PGD, MI or DI. When combined with TI, SGM also improves the TI attack by a considerable margin across all transfer scenarios. This indicates that, although manipulating the skip connections alone may not sufficient to attack secured models, it still can make existing attacks more powerful. One interesting observation is that attacks crafted here on RN152 are more transferable than those crafted on DN201, which is quite the opposite to attacking unsecured models.

## 4.3 A CLOSER LOOK AT SGM

In this part, we conduct more experiments to investigate the gradient decay factor of our proposed SGM, and explore the potential use of SGM for ensemble-based attacks and white-box attacks.

**Effect of Residual Gradient Decay $\gamma$.** We test the transferability of our proposed SGM with varying decay parameter $\gamma \in [0.1, 1.0]$, where $\gamma = 1.0$ means no decay on the residual gradients. The attacks are crafted by 10-step SGM on 5000 random ImageNet validation images. The results against 3 target models (VGG19, SE154 and IncV3) are illustrated in Figure 3. As can be observed, the trends are very consistent against different target models. On DenseNet source models, decreasing decay parameter (increasing decay strength) tends to improve transferability until it exceeds a certain threshold, *e.g.*, $\gamma = 0.5$. This is because the decay encourages the attack to focus on more transferable low-level information, however, it becomes less sufficient if all high-level class-relevant information is ignored. On ResNet source models, decreasing decay parameter can constantly improve transferability for $\gamma \geq 0.2$. Compared to DenseNet source models, ResNets require more

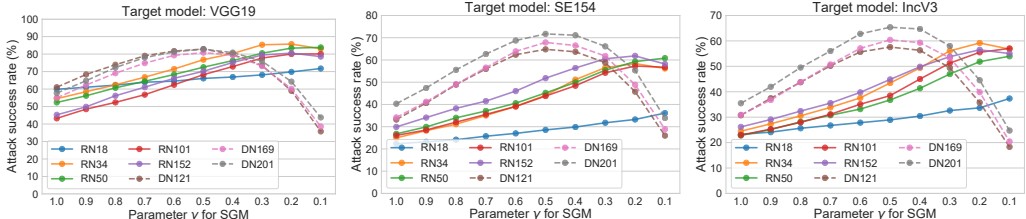

Figure 3: Parameter tuning: the success rates of black-box attacks crafted by 10-step SGM with varying decay parameter $\gamma \in [0.1, 1.0]$. The solid and dash curves represent results on ResNet and DenseNet source models respectively.

Table 6: Multi-step transferability of ensemble-based attack: the success rates (%±std over 5 random runs) of multi-step attacks crafted by different methods on an ensemble of 3 source models (*e.g.* RN34, RN152 and DN201) against 7 *unsecured* target models. The best results are in **bold**.

| Source | Attack | VGG19 | RN152 | DN201 | SE154 | IncV3 | IncV4 | IncRes |
|--------|--------|-------|-------|-------|-------|-------|-------|--------|
| RN34 + RN152 + DN201 | PGD | 86.69±0.20 | **99.99±0.01** | 99.99±0.02 | 69.65±0.71 | 65.95±0.35 | 59.30±0.32 | 53.91±0.40 |
| | TI | 84.35±0.21 | 99.59±0.11 | 99.77±0.06 | 71.67±0.19 | 67.22±0.82 | 66.02±0.66 | 56.83±0.89 |
| | MI | 92.86±0.19 | 99.91±0.04 | 99.91±0.06 | 86.11±0.38 | 83.25±0.35 | 79.25±1.16 | 76.53±0.66 |
| | DI | 96.34±0.23 | 99.84±0.20 | 99.84±0.20 | 89.72±0.52 | 87.53±0.29 | **85.04±0.75** | **81.11±0.44** |
| | **SGM** | **97.36±0.17** | 99.87±0.07 | 99.86±0.09 | **90.40±0.26** | **87.86±0.51** | 82.97±0.71 | 80.93±0.55 |
| | **DI+SGM** | **98.65±0.08** | 99.84±0.04 | 99.86±0.04 | **94.36±0.19** | **93.08±0.41** | **89.56±0.07** | **88.27±0.43** |

Table 7: Transferability of ensemble-based attack against secured models: the success rates (%±std over 5 random runs) of black-box attacks crafted on an ensemble of 3 source models (*e.g.* RN34, RN152 and DN201). The best results are in **bold**.

| Source | Attack | IncV3$_{ens3}$ | IncV$_{ens4}$ | IncRes$_{ens3}$ |
|--------|--------|----------------|---------------|-----------------|
| RN34 + RN152 + DN201 | PGD | 37.63±0.37 | 32.69±0.62 | 23.49±0.55 |
| | TI | **75.04±0.50** | **75.94±0.63** | **66.24±0.45** |
| | MI | 54.68±0.27 | 50.24±0.48 | 39.27±0.33 |
| | DI | 65.29±0.31 | 57.48±0.45 | 46.41±0.42 |
| | SGM | 66.08±0.42 | 62.22±0.73 | 51.16±0.12 |
| | **TI+SGM** | **87.65±1.00** | **85.11±0.27** | **77.75±0.41** |

decay on the residual gradients. Recalling that skip connections reveal more transferable information of the source model, ResNets require more penalty on the residual gradients to increase the importance of skip gradients that reveal more transferable information of the source model.

As for the selection of $\gamma$ under a scenario without knowing the target model, from Figure 3 and Appendix C, we can see that the influence of $\gamma$ is more related to the source model rather than the target model, that is, given a source model, the best $\gamma$ against different target models are generally the same. This makes the selection of $\gamma$ quite straightforward: choosing the best $\gamma$ on the source model(s). For instance, in Figure 3, suppose the unknown target model is SE154 (middle figure), the adversary could tune $\gamma$ on source model DN201 to attack VGG19 (left figure) and find the best $\gamma = 0.5$. The attacks crafted on DN201 with $\gamma = 0.5$ indeed achieved the best success rate against the SE154 target model (and other target models).

**Ensemble-based Attacks.** It has been shown that attacking multiple source models simultaneously can improve the transferability of the crafted adversarial examples, and is commonly adopted in practice. We follow the ensemble-based strategy (Liu et al., 2017) and craft attacks on an ensemble of RN34, RN152 and DN201. According to the discussion above, we select the best $\gamma$ individually for each source model: choose $\gamma$ for source RN34 and RN152 against target DN201, and $\gamma$ for source DN201 against target RN152. The success rates (transferability) against 7 unsecured models and 3 secured models are reported in Table 6 and Table 7 respectively. Similar to the results of single source model attacks, against unsecured target models, SGM has a similar standalone performance with DI, better than the others (except the two "white-box" scenarios against RN152 and DN201). When combined with other methods, *e.g.* DI, it improves the success rates again by a large margin.

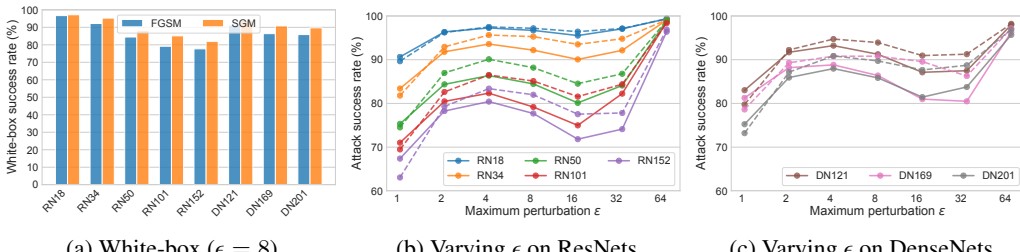

(a) White-box ($\epsilon = 8$)     (b) Varying $\epsilon$ on ResNets     (c) Varying $\epsilon$ on DenseNets

Figure 4: White-box success rate for FGSM versus SGM. In (b) and (c), each color corresponds to one model, with FGSM is represented by solid curve and SGM is represented by dashed curve.

Against secured models, SGM achieves the second best standalone transferability, with TI is still the best. When combined with TI, SGM improves the success rate by $\sim 10\%$ consistently against all secured target models. In particular, against IncV3$_{ens3}$, TI+SGM achieves higher success rate (87.65%) than reported in (Dong et al., 2019) (84.8%), even if only 3 source models are used here and the source models (*e.g.* RN34, RN152 and DN201) are all of different architecture to IncV3 target model ((Dong et al., 2019) uses 6 source models including even the IncV3 model). From all aspects analyzed above, the existence of skip connections makes transferable attacks much easier to craft in practice.

**Improving Weak White-box Attacks.** In addition to the black-box transferability, we next show that SGM can also improve the weak (one-step) white-box attack FGSM. Note that the one-step version of SGM is equivalent to FGSM plus residual gradient decay. Our experiments are conducted on the 8 source models, and the white-box success rates under maximum $L_\infty$ perturbation $\epsilon = 8$ (a typical white-box setting) are shown in Figure 4a. As can be observed, using SGM can help improve the adversarial strength (*i.e.*, higher success rate). We then vary the maximum perturbation $\epsilon \in [1, 64]$, and show the results on ResNet and DenseNet models separately in Figure 4b and Figure 4c. Compared to FGSM, SGM can always give better adversarial strength, except when $\epsilon$ is extremely small ($\epsilon \leq 2$). When the perturbation space becomes infinitely small, the loss landscape within the space becomes flat and the gradient points to the optimal perturbation direction. However, when the perturbation space expands, one-step gradient becomes less accurate due to changes in the loss landscape (success rate decreases as $\epsilon$ increases from 4 to 16), and in this case, the skip gradient which contains more low-level information is more reliable than the residual gradient (the improvement is more significant for $\epsilon \in [4, 16]$ ). Another interesting observation is that adversarial strength decreases when the model becomes more complex from RN18 to RN152, or DN121 to DN201. This is likely because the loss landscape of complex models is steeper than shallow models, making one-step gradient less reliable.

## 5   CONCLUSION

In this paper, we have identified a surprising property of the skip connections used by many state-of-the-art ResNet-like neural networks, that is, they can be easily used to generate highly transferable adversarial examples. To demonstrate this architectural "weakness", we proposed the *Skip Gradient Method* (SGM) to craft adversarial examples using more gradients from the skip connections rather than the residual ones, via a decay factor on gradients. We conducted a series of transfer attack experiments with 8 source models and 10 target models including 7 unsecured and 3 secured models, and showed that attacks crafted by SGM have significantly better transferability than those crafted by existing methods. When combined with existing techniques, SGM can also boost state-of-the-art transferability by a huge margin. We believe the high adversarial transferability of skip connections is due to the fact that they expose extra low-level information which is more transferable across different DNNs. Our findings in this paper not only remind researchers in adversarial research to pay attention to the architectural vulnerability of DNNs, but also raise new challenges for secure DNN architecture design.

## ACKNOWLEDGEMENT

Shu-Tao Xia is supported in part by National Key Research and Development Program of China under Grant 2018YFB1800204, National Natural Science Foundation of China under Grant 61771273, R&D Program of Shenzhen under Grant JCYJ20180508152204044, and research fund of PCL Future Regional Network Facilities for Large-scale Experiments and Applications (PCL2018KP001).

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

## A   VISUALIZATION OF ADVERSARIAL EXAMPLES CRAFTED BY SGM

In this section, we visualize 6 clean images and their corresponding adversarial examples crafted using our SGM on either a ResNet-152 or a DenseNet201 in Figure 5. These visualization results show that the generated adversarial perturbations are human imperceptible.

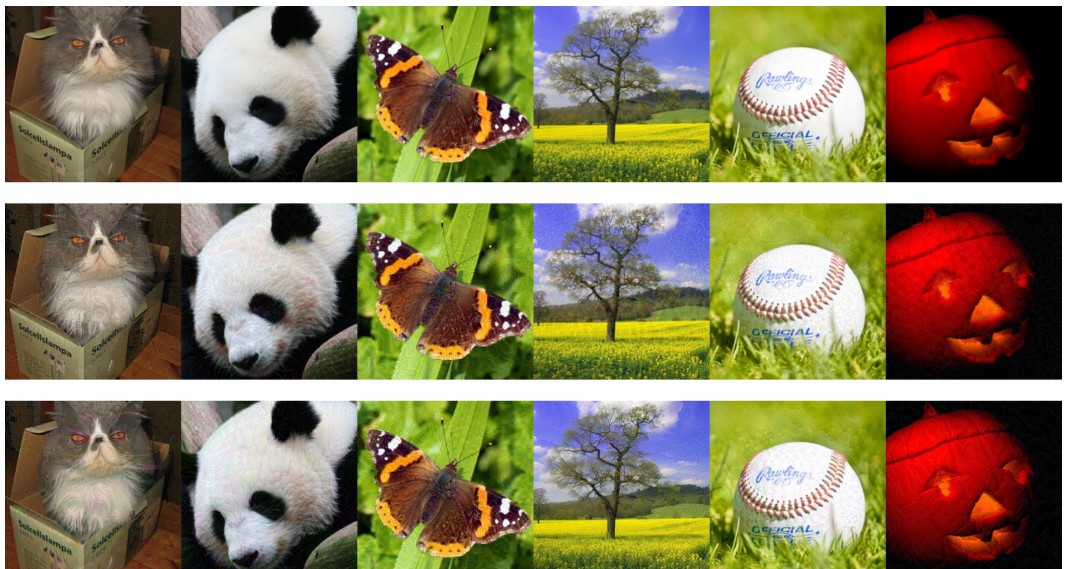

Figure 5: Visualization of 6 clean images and their corresponding adversarial examples. The clean images are shown in the top row, adversarial images crafted on ResNet-152 are shown in the middle row, while those crafted on DenseNet-201 are shown in the bottom row. All adversarial images are crafted using our proposed SGM (10-step) under maximum perturbation $\epsilon = 16$.

## B   COMPARISON WITH PREVIOUSLY PUBLISHED RESULTS

In this section, we compare the experimental settings in previous and our works, and discuss some small discrepancies of the baseline performance reported in ours and previous works.

Table 8 and 9 summarizes these differences for single-source and ensemble-based attack respectively. Out of all these works (Dong et al., 2018; 2019; Xie et al., 2019), results reported in (Xie et al., 2019) are more complete. Our reported success rate of baseline attacks (*e.g.* MI and DI) matches that reported in (Xie et al., 2019), sometimes even higher. The slight discrepancy is caused by the difference in experimental settings. Table 10 summarizes the different source models used by baseline attacks, and Table 11 summarizes the difference in dataset, number of test images, input image size, maximum $L_\infty$ perturbation $\epsilon$, number of attack steps $N$ and attack step size $\alpha$. Compared to $299 \times 299$ image size, here we use a more standard image size $224 \times 224$ on ImageNet. The use of smaller input size may reduce the effectiveness of existing attacks (Simon-Gabriel et al., 2019).

In another work by Liu et al. (2017), 81% success rate was reported for optimization-based attack crafted on ResNet-152 against target VGG16, which is higher than our 65.52% from ResNet-152 to VGG19. This is because they did not restrict the maximum perturbation $\epsilon$. The *root mean square deviation* (RMSD) of their attacks is 22.83,which indicates that many pixels are perturbed more than 16 pixel values. In our experiments, the RMSD is 6.29 for PGD, 7.71 for SGM, and 12.55 for MI. This appears to be another reason for the performance discrepancy. Note that the advantage of bounded small perturbation is increasing imperceptibility to human observers (see Figure 5).

Table 8: Previously reported attack success rates (%) of baseline single-source attacks against 6 target models. "-" means no results were reported.

| Reference | Attack | IncV3 | IncV4 | IncRes | $IncV3_{ens3}$ | $IncV3_{ens4}$ | $IncRes_{ens3}$ |
|---|---|---|---|---|---|---|---|
| Dong et al. (2018) | FGSM | 35.0 | 28.2 | 27.5 | 14.6 | 13.2 | 7.5 |
| | BIM | 26.7 | 22.7 | 21.2 | 9.3 | 8.9 | 6.2 |
| | MI | 53.6 | 48.9 | 44.7 | 22.1 | 21.7 | 12.9 |
| Dong et al. (2019) | FGSM | - | - | - | 20.2 | 17.7 | 9.9 |
| | MI | - | - | - | 25.1 | 23.7 | 13.3 |
| | DI | - | - | - | 40.5 | 36.0 | 24.1 |
| Xie et al. (2019) | FGSM | 34.4 | 28.50 | 27.1 | 12.4 | 11.0 | 6.0 |
| | PGD | 20.8 | 17.2 | 14.9 | 5.4 | 4.6 | 2.8 |
| | MI | 50.1 | 44.1 | 42.2 | 18.2 | 15.2 | 9.0 |
| | DI | 53.8 | 49.0 | 44.8 | 13.0 | 11.1 | 6.9 |
| Ours | FGSM | 26.56 | 21.03 | 19.10 | - | - | - |
| | PGD | 26.56 | 21.03 | 19.10 | 12.47 | 10.72 | 6.97 |
| | MI | 50.22 | 43.32 | 41.71 | 24.20 | 22.04 | 16.10 |
| | DI | 53.95 | 47.16 | 43.47 | 34.84 | 29.23 | 21.64 |

Table 9: Previously reported attack success rates (%) of ensemble-based baseline attacks against 6 target models. "-" means no results were reported.

| Reference | Attack | IncV3 | IncV4 | IncRes | $IncV3_{ens3}$ | $IncV3_{ens4}$ | $IncRes_{ens3}$ |
|---|---|---|---|---|---|---|---|
| Dong et al. (2018) | FGSM | 45.7 | 39.9 | 38.8 | 15.4 | 15.0 | 6.4 |
| | BIM | 72.1 | 61.0 | 54.4 | 18.6 | 18.7 | 9.9 |
| | MI | 87.9 | 81.2 | 76.5 | 37.6 | 40.3 | 23.3 |
| Dong et al. (2019) | FGSM | - | - | - | 27.5 | 23.7 | 13.4 |
| | MI | - | - | - | 50.5 | 48.3 | 32.8 |
| | DI | - | - | - | 66.0 | 63.3 | 45.9 |
| | TI+DI | - | - | - | 84.8 | 82.7 | 78.0 |
| Xie et al. (2019) | PGD | 43.7 | 36.4 | 33.3 | 12.9 | 15.1 | 8.8 |
| | MI | 69.9 | 67.9 | 64.1 | 36.3 | 35.0 | 30.4 |
| | DI | 71.4 | 65.9 | 64.6 | 22.8 | 26.1 | 15.8 |
| | DI+MI | 80.7 | 80.6 | 80.7 | 44.6 | 44.5 | 39.4 |
| Ours | PGD | 65.95 | 59.30 | 53.91 | 37.63 | 32.69 | 23.49 |
| | MI | 83.25 | 79.25 | 76.53 | 54.68 | 50.24 | 39.27 |
| | DI | 87.53 | 85.04 | 81.11 | 65.29 | 57.48 | 46.41 |
| | DI+SGM | 93.08 | 89.56 | 88.27 | 80.14 | 76.52 | 66.40 |
| | TI+SGM | - | - | - | 87.65 | 85.11 | 77.75 |

Table 10: Source models used by existing single-source and ensemble-based black-box attacks. "Hold-out" refers to the hold-out target model from the group, with all remaining models are used as source models. Group 1 consists of ResNet-v2-152, IncV3, IncV4 and IncRes, while group 2 consists of ResNet-v2-152, IncV3, IncV4, IncRes, $IncV3_{ens3}$, $IncV3_{ens4}$ and $IncRes_{ens3}$.

| Reference | Single-source attack | Ensemble-based attack |
|---|---|---|
| Dong et al. (2018) | ResNet-v2-152 | hold-out from group 1 or group 2 |
| Dong et al. (2019) | ResNet-v2-152 | hold-out from group 1 |
| Xie et al. (2019) | ResNet-v2-152 | hold-out from group 2 |
| Ours | RN152 | RN34 + RN152 + DN201 |

For proper implementation, we use open-source codes and pretrained models for our experiments, *e.g.*, AdverTorch (Ding et al., 2019) for FGSM, PGD and MI, and source/target models from two GitHub repositories[†‡§] for all models. We reproduced DI and TI in PyTorch.

---

[†] https://github.com/Cadene/pretrained-models.pytorch

[‡] https://github.com/tensorflow/models/tree/master/research/slim

[§] https://github.com/tensorflow/models/tree/master/research/adv_imagenet_models

Table 11: Difference in experimental settings of our work compared to previous works."NeurIPS 2017" indicates the dataset used for NeurIPS 2017 adversarial competition. $\epsilon$: maximum per-pixel perturbation; $N$: number of attack steps; $\alpha$: attack step size.

| Reference | Dataset | Number | Input size | $\epsilon$ | $N$ | $\alpha$ |
|---|---|---|---|---|---|---|
| Dong et al. (2018) | ImageNet | 1000 | $299 \times 299$ | 16 | 10 | 1.6 |
| Dong et al. (2019) | NeurIPS 2017 | 1000 | $299 \times 299$ | 16 | 10 | 1.6 |
| Xie et al. (2019) | ImageNet | 5000 | $299 \times 299$ | 15 | 19 | 1.0 |
| Ours | ImageNet | 5000 | $224 \times 224$ | 16 | 10 / 20 | 2.0 |

## C   TRANSFERABILITY OF DECAY PARAMETER $\gamma$

In this section, we study the "transferability" of decay parameter $\gamma$ across different target models. RN152 and DN201 are used as the source model, and the target model are varying to observe trends. As indicated in Figure 6, all black-box target models share the same best selection of $\gamma$, which makes $\gamma$ selection quite simple. Even if the true target model is unknown, the adversary can tune the gamma for a resnet-like neural network through another model and obtain the best selection as well.

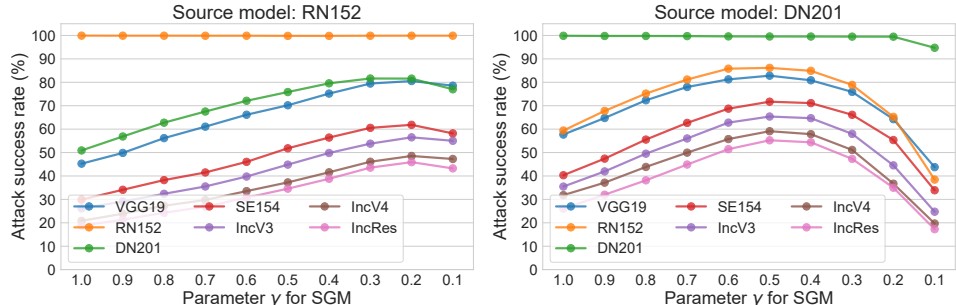

Figure 6: "Transferability" of decay parameter: the success rates of black-box attacks crafted by 10-step SGM with varying decay parameter $\gamma \in [0.1, 1.0]$. The curves represent results against different target models respectively.

