# OpenReview forum: "Skip Connections Matter: On the Transferability of Adversarial Examples Generated with ResNets"
_ICLR.cc/2020/Conference — Accept (Spotlight)_

### Official Review · AnonReviewer3 · 2019-10-21
**Official Blind Review #3**

**Rating:** 6

**Review:**

The paper is about adversarial attacks and highlights a security weakness of skip connections in ResNet-like CNNs, namely: skip connections make it easier to obtain adversarial examples. This observation leads to new approach to adversarial attacks, named Skip Gradient Method (SGM), which weights the residual gradient w.r.t  the skip connection gradient. The approach is validated on a variety of image classification attack scenarios (e. g. white-box and transfer attacks) using two families of source models (ResNet and DenseNet). The results show the superiority of SGM when comparing to other adversarial attack scenarios.

Strengths:
- Simple approach that seems to be giving good results
- Large number of adversarial attack scenarios tested
- Good related work review

Weaknesses:
- Results are reported without variance information
- There are some details missing on how the decay factor is selected
- Results are reported only on one dataset (ImageNet)

The paper is well written, the authors have identified a "problem" of ResNet-like models and proposed an approach that can exploit the problem in adversarial attacks scenarios (SGM). To the best of my knowledge, this is the first time someone has identified the skip connections security problem in ResNets. The SGM is compared against many existing adversarial attacks methods achieving making the evaluation section quite detailed. Thus, I'd lean towards paper acceptance.

However, I have some questions with respect to the evaluation section:
1. Although, the paper includes ablation analysis for different values of the decay factor, I could not find details on how the decay factor hyperparamenter is selected. Given that this factor is a hyperparamenter, it feels like it should be selected on a validation set and tested on a test set. The paper comments about 5000 random validation set images that are used to compere methods (test set), however, I could not find any mention about validation set used to select decay factor. Could the authors specify how the decay factor is set?
2. Since all the results are reported with 5000 random imagent images, it would be interesting to see the variance of results if the sampling of images is repeated.
3. Section 3.3, 2nd paragraph: "Another important observation is that when there are more skip connections in a network.... the crafted attacks become more transferable....". Although, this statement seems to be correct when comparing DenseNet to ResNet, it does not necessarily hold when comparing models within the same family (e. g. RN18 to RN 152 for FGSM or RN34 to RN 152 for PGD) suggesting that the best is not only connected to the number of skip connections. Maybe the superiority of DN is rather related to the nature of the DN models and not to the number of skip connections. Could the authors clarify?
4. Section 4, threat models: "... the same architectures but trained separately." Could the authors clarify what does it mean? Are these models re-trained from scratch changing the random seed of model initialization, order of the dataset, or something else? In general are the authors using pertained models or train all the models "from scratch".
5. Fig 2: Since x-axis values are not continuous - it might be better to use bar plot.
6. In some tables, the results are reported just for SGM while in other tables SGM is reported in combination with other method. I guess that when the results are reported for SGM they represent SGM from Eq. 10 that represents SGM+PGD. Is this correct? The authors might consider clarifying this in the paper.

Some typos:
In the following, we exploits an architectural security weakness about....
... the the crafted attacks...
... not only reminds researcher to pay....


**Experience Assessment:**

I have read many papers in this area.

**Review Assessment: Checking Correctness Of Derivations And Theory:**

N/A

**Review Assessment: Checking Correctness Of Experiments:**

I assessed the sensibility of the experiments.

**Review Assessment: Thoroughness In Paper Reading:**

I read the paper at least twice and used my best judgement in assessing the paper.

---

> ### Author Response · Authors · 2019-11-13
> **Response to Reviewer #3**
>
> Thanks for your careful reading and valuable suggestions for improvements.
>
> Q1: Results are reported without variance information.
>
> A1: Thanks for your suggestion. We have added the variance information in Tables 1-3 and Table 5-7.
>
> Q2: There are some details missing on how the decay factor is selected.
>
> A2: We have added a new subsection 4.4 to discuss the selection of decay factor γ in practice, and an additional study on the "transferability" of $\gamma$ in Appendix C. As can be seen from our parameter tuning in Figure 3 and Figure 7, $\gamma$ is more associated with the source model rather than the target model. The "transferability" of $\gamma$ is quite good and stable. For example, given source model DenseNet-201, the highest success rate is always achieved at $\gamma=0.5$ against all target models such as VGG19, SE154 or Inception-V3. In other words, the selection of $\gamma$ is quite simple and straightforward: tune $\gamma$ on the source model (which is known) against some random target model.
>
> Q3: Results are reported only on one dataset (ImageNet).
>
> A3: The main reason is that many recent studies on transferable adversarial attacks are based only on ImageNet [1][2][3]. For straightforward and fair comparisons, our experiments were also conducted on ImageNet.
>
> [1] Yinpeng, Dong, et al. Boosting adversarial attacks with momentum. CVPR, 2018.
> [2] Yinpeng, Dong, et al. Improving Black-box Adversarial Attacks with a Transfer-based Prior. CVPR, 2019.
> [3] Xie, Cihang, et al. Improving transferability of adversarial examples with input diversity. CVPR. 2019.
>
> For questions with respect to the evaluation section, we address them in detail below.
>
> Q4.1: Could the authors specify how the decay factor is set?
>
> A4.1: Yes, hyper-parameters are typically selected based on a validation set. However, in this adversarial attack setting, the adversary would exploit all the data at hand to craft the strongest adversarial examples, which means there is no need to split data into val and test. The real challenge here is how to select the optimal decay parameter so as to craft the strongest attacks when the target model is unknown. We have addressed this question in the above Q2&A2, and added a discussion on the selection of the decay factor in Section 4.4.
>
> Q4.2: Since all the results are reported with 5000 random imagent images, it would be interesting to see the variance of results if the sampling of images is repeated.
>
> A4.2: We have added variance information in Tables 1-7 over 5 repetitions of the experiments with different random seeds.
>
> Q4.3: Section 3.3, 2nd paragraph: "Another important observation is that when there are more skip connections in a network.... the crafted attacks become more transferable....". This statement seems to be correct when comparing DenseNet to ResNet, it does not necessarily hold when comparing models within the same family.
>
> A4.3: We would like to clarify that the RN18/RN34 are of quite different architectures compared to RN50/RN101/RN152. Specifically, RN18/RN34 are built with normal residual blocks, while RN50/RN101/RN152 are built with “bottleneck” residual blocks. We have added discussion to clarify this and adjusted our claims accordingly in the last paragraph of Section 3.3.
>
> Q4.4: Section 4, threat models: "... the same architectures but trained separately." Could the authors clarify what does it mean?
>
> A4.4: Thanks for pointing this out, we made an error here. The correct description is “The target model is of a different architecture (indicated by the model name) to the source model, except when the source and target models are of the same architecture, where we directly use the source model as the target model (equivalent to a white-box setting).” We have fixed this in the revision. A detailed description of experimental settings has also been added to Appendix B. Please note that our discussion on experimental results is still correct,  e.g., “when the source and target models are the same, the result represents the white-box success rate.”
>
> Q4.5: Since x-axis values are not continuous - it might be better to use bar plot.
>
> A4.5: Thanks for the suggestion. The choice of curve plots over bar plots is for clarity purposes. We find the bar plots are less clear than the curve plots, as there will be too many bars crowded in one figure: 5 bars at each x-value and 40 bars in a single figure.
>
> Q4.6: In some tables, the results are reported for SGM while in other tables SGM is reported in combination with other methods.
>
> Q4.6: Thanks for pointing this out. We have unified and clarified all the notations to SGM in the revision. In Section 3.3 Table 1, SGM refers to PGD+SGM. In experiment Section 4, for single-step attack, SGM is equivalent to FGSM+SGM, while for multi-step attacks, SGM is equivalent to PGD+SGM.

---

### Official Review · AnonReviewer4 · 2019-10-25
**Official Blind Review #4**

**Rating:** 8

**Review:**

Summary:
This paper proposes a modification to standard Projected Gradient Descent to improve transferability of adversarial examples, when the source model is a ResNet-like model containing skip connections. The method, Skip Gradient Method (SGM) modifies the backwards pass to scale down the gradient computed in each residual branch of the model, before these gradients are combined with the gradient from the skip connection. This thus upweights the gradients from the skip connections as opposed to residual modules. The paper demonstrates significant improvements in the single-model black-box transfer setting, against a variety of undefended and defended models.

Strengths:
- Lots of interesting empirical results here! One result which stands out to me is Table 4, where across a wide range of target models, adding SGM to existing techniques cuts defender accuracy by ~1/2 (e.g. 79.9% to 89.66% for SE154). The results in Table 3, showing that even without additional techniques, SGM results in large improvements and outperforms previous approaches, are also quite nice.
- It's notable that such a simple approach leads to significant improvements.
- Results are very clearly presented, and writing is clear throughout.

Suggestions for improvement:

I have 3 major concerns: (1) discrepancies between baselines and previously published results (2) unrealistic threat model (3) framing / conclusions drawn by paper.

I suspect (1) is easily addressed, but was not clear to me from the current paper.

(1) Baselines:
- For multi-step transfer against undefended models (Table 3), the attack success rates seem low compared to numbers reported in e.g. Liu et al. For example, using ResNet-152 as the source, and VGG-19 as the target, Liu et al reports 19% defender accuracy = 81% attacker success. This is significantly higher than the 65.52% reported for MI (both are non-regularized optimization attacks), also stronger than the 80.68% reported for SGM. In general, considering these are *untargeted* attacks, with eps=16, the transfer rates seem pretty low.
- For multi-step transfer against defended models (Table 5), the numbers are slightly lower than previously reported. E.g. attacking IncV3_ens3, Dong et al 19 reports 46.9% for MI, but here, 44.28% is reported. (I realize these differences are slight.)
- In general, it would be nice if the paper were structured so as to make these comparisons easier. For example, the appendix could include tables comparing the baselines reported here, to values previously reported, and explain any discrepancies. Particularly with black-box transfer, where baseline performance is so sensitive to small choices, it's important to ensure baselines are properly implemented, and the current writing of the paper makes it impossible for the reader to assess this unless they are very familiar with the black-box transfer literature.

(2) Unrealistic threat model:
- For black-box transfer, all the strongest attacks use multiple source models (see e.g. NeurIPS 2017 Adversarial Examples contest, the baselines cited in the paper). While this paper shows significant improvements in the single-source setting, these results are significantly weaker than any multiple-source attack. For instance, Liu et al 17 achieve near 100% untargeted success (and near 100% targeted success) against all undefended models they study (many which overlap), and Dong et al 19 report ~85% accuracy against IncV3_ens3 (compared to <60% here).
- There's no reason in practice that an adversary would not employ an ensemble-based attack if they wanted to fool an unknown model.

(3) Framing / conclusions:
- The paper frames the results as a "security vulnerability of ResNets," but the results don't show this. In particular, they show that ResNets make effective *source* models to be used by *attackers*, but they don't imply which models defenders should use in order to be robust to black-box attacks. In this way, the main message of the paper seems misleading to practitioners.
- The main message of the paper thus seems to be "on the transferability of adversarial examples generated  with resnets" as opposed to the "security of skip connections." I would encourage rewriting of the title/intro/conclusion as such.

Overall, there are several very interesting empirical findings in this paper. I view the two main impacts these results could later have would be leading to (1) improved understanding of what causes transfer of adversarial examples and (2) improved understanding of ResNet-like architectures (for example, the results provide some support for the view of ResNets as ensembles of shallow models, cited in the paper). If the paper were written with this view, then concern (2) above becomes unimportant.


Suggestions I believe could strengthen the paper, but I do not view as weaknesses, or necessities:
- Can you decouple the effect of SGM on optimization and transfer? The paper does a bit of this, but restricts the analysis to the single-step case. For instance, if the main effect of SGM is on optimization, this has interesting implications for optimization of ResNets. If the main effect is on transfer, this has interesting implications for what causes transfer.

Minor:
- I'd suggest moving Table 1 (one-step attacks) to the Appendix. The results are less impressive than multi-step, and the community as a whole favors stronger attacks.

Overall, there are several interesting empirical findings in this paper (modulo concerns about baselines indicated above). I suggest that the paper either consider more realistic threat models to be useful to the adversarial examples field, or focus on the insights revealed by these findings. I hope that the authors can address the concerns outlined here, and I would be happy to adjust my score if so.

Note that my current indicated confidence rating is for the current state of the paper. I would be happy to adjust my evaluation if revisions to the paper can one/several of the concerns indicated above.


__________________

EDIT: Rating adjusted from 3: Weak Reject to 8: Accept, after accounting for revisions and author rebuttal. See reply below.

**Experience Assessment:**

I have published in this field for several years.

**Review Assessment: Checking Correctness Of Derivations And Theory:**

N/A

**Review Assessment: Checking Correctness Of Experiments:**

I carefully checked the experiments.

**Review Assessment: Thoroughness In Paper Reading:**

I read the paper thoroughly.

---

> ### Author Response · Authors · 2019-11-13
> **Response to Reviewer #4**
>
> Thank you for your valuable and thoughtful comments.
>
> Q1: The discrepancies between baselines and previously published results.
>
> A1: As suggested, we have added four tables and detailed discussions in Appendix B: 1）Table 8 compares the reported difference in attack success rate of single-source baseline attacks; 2) Table 9 compares the difference on ensemble-based baseline attacks; 3) Table 10 summarizes the different source models used by baseline and our attacks; and 4) Table 11 summarizes the difference in experimental settings (number of test examples, input size, maximum perturbation $\epsilon$ etc.). Overall, our reported success rate of baseline attacks matches that reported in one of the more complete works done in [1], sometimes even higher. The slight discrepancy is caused by the difference in experimental settings:
> - The attack of Liu et al. 2017 does not restrict the maximum perturbation $\epsilon$. The root mean square deviation (RMSD) of their crafted adversarial examples is 22.83, which means many pixels are perturbed more than 16 pixel values. In our experiments, the RMSD is 6.29 for PGD, 7.71 for SGM, and 12.55 for MI. As indicated in Huang et al., 2019, black-box attacks with larger perturbations are generally more effective. Thus, it is not a surprise that success rates reported in Liu et al. 2017 are higher than ours.
>
> - In Dong et al (2019), 46.9% is the success rate for TI+DI, while 44.28% in our paper is the result for TI only.
>
> [1] Xie, Cihang, et al. Improving transferability of adversarial examples with input diversity. CVPR. 2019.
>
> Q2: Considering ensemble threat model.
>
> A2: Following the suggestions, we have run additional experiments using ensemble-based attacks and reported the results in Section 4.4. The result indicates that our proposed SGM or a combination of SGM with other existing methods is indeed a good choice for crafting more powerful attacks. In particular, using an ensemble of only 3 source models (ResNet-34, ResNet-152, and DenseNet-201) with TI+SGM, we are able to achieve 87.65% success rate against IncV3_ens3 target model, surpassing the 84.8% success rate reported in Dong et al, (2019) using an ensemble of 6 models with TI+DI. Detailed discussions can be found in Section 4.4.
>
> Q3: Alternative paper title and rephrasing of claims.
>
> A3: We agree with the two different points of views proposed by the reviewer. However, we also believe it is important to emphasize this surprising property of skip connections in contrast to their power for constructing very deep neural networks. Therefore, we have modified the title to “Skip Connections Matter: On the Transferability of Adversarial Examples Generated with ResNets” and have adjusted the title/intro/conclusion accordingly.
>
> Q4: Can you decouple the effect of SGM on optimization and transfer?
>
> A4: Thanks for your insightful comments. We will explore and study how to decouple the effect of SGM on optimization and transferability in our further work.
>
> Q5: Move Table 1 (one-step attacks) to the Appendix.
>
> A5: We agree that the community as a whole favors stronger attacks, and have removed results of FGSM in Table 1.

---

> > ### Comment · AnonReviewer4 · 2019-11-13
> > **Thank you for the significant additional experiments, responses, and presentation revisions.**
> >
> > Thank you for the significant additional experiments, responses, and presentation revisions. These address my previous concerns, and I believe greatly strengthen the reliability and significance of the overall message. I believe this paper contains several interesting empirical results, which will be of great interest to the ICLR community (in particular, to those interested in black-box adversarial transfer, and the nature of representations learned by ResNets).
> >
> > I have adjusted by overall rating above, from a Weak Reject (3) to Accept (8).

---

### Official Review · AnonReviewer2 · 2019-10-28
**Official Blind Review #2**

**Rating:** 6

**Review:**

The paper discovers a very interesting phenomenon that adversarial examples are more transferable when the perturbations are obtained by propagating a higher ratio of gradients via shortcuts in ResNets and DenseNets.
The method is simple and comprehensive experiments are conducted to prove its correctness.
It is a good empirical paper that can inspire future research on investigating the role of shortcuts when defending adversarial examples.

To improve the paper:
1. The "transferability" (or robustness) of gamma γ should be studied. In real applications, γ should be selected before transferring to new black-box models. The experiments in this paper are done by reporting the best results under the best γ. This is a hindsight. How can you pick up γ without accessing the transferability results?
2. More theoretical explanations. It is unclear to me that why it works so well, since the back propagation is no longer coherent with the forward function.

**Experience Assessment:**

I have published one or two papers in this area.

**Review Assessment: Checking Correctness Of Derivations And Theory:**

N/A

**Review Assessment: Checking Correctness Of Experiments:**

I carefully checked the experiments.

**Review Assessment: Thoroughness In Paper Reading:**

I read the paper thoroughly.

---

> ### Author Response · Authors · 2019-11-13
> **Response to Reviewer #2**
>
> Thank you for your insightful suggestions for improving the paper.
>
> Q1: How to pick up $\gamma$ without accessing the transferability results?
>
> A1: We have added a new subsection 4.4 to discuss the selection of  $\gamma$ in practice, and an additional study on the "transferability" of $\gamma$ in Appendix C. As can be seen from our parameter tuning in Figure 3 and Figure 7, $\gamma$ is more associated with the source model than the target model. The "transferability" of $\gamma$ is quite good and stable. For example, given source model DenseNet-201, the highest success rate is always achieved at $\gamma=0.5$ against all target models such as VGG19, SE154 or Inception-V3. In other words, the selection of $\gamma$ is simple and straightforward: tune $\gamma$ on the source model (which is known) against some random target model.
>
> Q2: Why it works so well, since the back propagation is no longer coherent with the forward function.
>
> A2:  At a high-level, the use of SGM exploits more low-level features of the source model (via gradient backpropagation). Low-level features are more transferable to target models because different DNNs tend to learn similar low-level features (forward function). We acknowledge that a more theoretical analysis can help understanding of our method, and will definitely consider this as our future work.

---

### Public Comment · ~Dinghuai_Zhang1 · 2020-03-06
**What's the difference with Ghost network?**

reference: Learning Transferable Adversarial Examples via Ghost Networks https://arxiv.org/abs/1812.03413.
seems similar?

---

> ### Author Response · Authors · 2020-03-06
> **Difference to Ghost network**
>
> Hi Dinghuai,
>
> Thanks for the reference. It appears that we have missed this one. The difference between our work and the Ghost Networks lies in the different usage of skip connections. Ghost Networks use skip connections with uniform noise in the forward process to sample more networks for the ensemble attack, while we study the architectural property of skip connections from the perspective of backpropagate gradient without modifying the forward process. In short, our approach demonstrates that you do not need to sample many networks, rather, a simple decay parameter on the gradients can do the job or even better. We believe the two works could be connected in some way, for which we believe is an interesting future work. We have uploaded a revision to clarify this in the related work.

---

### Decision · Program_Chairs · 2019-12-19

**Decision:**

Accept (Spotlight)

**Comment:**

This paper makes the observation that, by adjusting the ratio of gradients from skip connections and residual connections in ResNet-family networks in a projected gradient descent attack (that is, upweighting the contribution of the skip connection gradient), one can obtain more transferable adversarial examples. This is evaluated empirically in the single-model black box transfer setting, against a wide range of models, both with and without countermeasures.

Reviewers praised the novelty and simplicity of the method, the breadth of empirical results, and the review of related work. Concerns were raised regarding a lack of variance reporting, strength of the baselines vs. numbers reported in the literature,  and the lack of consideration paid to the threat model under which an adversary employs an ensemble of source models, as well as the framing given by the original title and abstract. All of these appear to have been satisfactorily addressed, in a fine example of what ICLR's review & revision process can yield. It is therefore my pleasure to recommend acceptance.